# Increased Galectin-9 Levels Correlate with Disease Activity in Patients with DMARD-Naïve Rheumatoid Arthritis and Modulate the Secretion of MCP-1 and IL-6 from Synovial Fibroblasts

**DOI:** 10.3390/cells12020327

**Published:** 2023-01-15

**Authors:** Morten A. Nielsen, Ditte Køster, Akul Y. Mehta, Kristian Stengaard-Pedersen, Pierre Busson, Peter Junker, Kim Hørslev-Petersen, Merete Lund Hetland, Mikkel Østergaard, Malene Hvid, Hakon Leffler, Tue W. Kragstrup, Richard D. Cummings, Bent Deleuran

**Affiliations:** 1Department of Biomedicine, Aarhus University, 8000 Aarhus, Denmark; 2Department of Rheumatology, Aarhus University Hospital, 8000 Aarhus, Denmark; 3Department of Surgery, Beth Israel Deaconess Medical Center, Harvard Medical School, Boston, MA 02115, USA; 4CNRS-UMR 9018-METSY, Université Paris-Saclay, Gustave Roussy, 94800 Villejuif, France; 5Department of Rheumatology, Odense University Hospital, 5000 Odense, Denmark; 6Danish Hospital for Rheumatic Diseases, University of Southern Denmark, 5230 Odense, Denmark; 7Copenhagen Center for Arthritis Research, Center for Rheumatology and Spine Diseases, Rigshospitalet, 2600 Glostrup, Denmark; 8Department of Clinical Medicine, University of Copenhagen, 2100 Copenhagen, Denmark; 9Department of Clinical Medicine, Aarhus University, 8000 Aarhus, Denmark; 10Division for Microbiology, Immunology and Glycobiology (MIG), Department of Laboratory Medicine, Lund University, 221 00 Lund, Sweden

**Keywords:** fibroblast, Galectin-9, inflammation, rheumatoid arthritis

## Abstract

**Background**: Fibroblast-like synoviocytes (FLSs) are essential mediators in the expansive growth and invasiveness of rheumatoid synovitis, and patients with a fibroblastic-rich pauci-immune pathotype respond poorly to currently approved antirheumatic drugs. Galectin-9 (Gal-9) has been reported to directly modulate rheumatoid arthritis (RA) FLSs and to hold both pro- and anti-inflammatory properties. The objective of this study was to evaluate clinical and pathogenic aspects of Gal-9 in RA, combining national patient cohorts and cellular models. **Methods**: Soluble Gal-9 was measured in plasma from patients with newly diagnosed, treatment-naïve RA (*n* = 98). The disease activity score 28-joint count C-reactive protein (DAS28CRP) and total Sharp score were used to evaluate the disease course serially over a two-year period. Plasma and synovial fluid samples were examined for soluble Gal-9 in patients with established RA (*n* = 18). A protein array was established to identify Gal-9 binding partners in the extracellular matrix (ECM). Synovial fluid mononuclear cells (SFMCs), harvested from RA patients, were used to obtain synovial-fluid derived FLSs (SF-FLSs) (*n* = 7). FLSs from patients suffering from knee Osteoarthritis (OA) were collected from patients when undergoing joint replacement surgery (*n* = 5). Monocultures of SF-FLSs (*n* = 6) and autologous co-cultures of SF-FLSs and peripheral blood mononuclear cells (PBMCs) were cultured with and without a neutralizing anti-Gal-9 antibody (*n* = 7). The mono- and co-cultures were subsequently analyzed by flow cytometry, MTT assay, and ELISA. **Results**: Patients with early and established RA had persistently increased plasma levels of Gal-9 compared with healthy controls (HC). The plasma levels of Gal-9 were associated with disease activity and remained unaffected when adding a TNF-inhibitor to their standard treatment. Gal-9 levels were elevated in the synovial fluid of established RA patients with advanced disease, compared with corresponding plasma samples. Gal-9 adhered to fibronectin, laminin and thrombospondin, while not to interstitial collagens in the ECM protein array. In vitro, a neutralizing Gal-9 antibody decreased MCP-1 and IL-6 production from both RA FLSs and OA FLSs. In co-cultures of autologous RA FLSs and PBMCs, the neutralization of Gal-9 also decreased MCP-1 and IL-6 production, without affecting the proportion of inflammatory FLSs. **Conclusions**: In RA, pretreatment plasma Gal-9 levels in early RA were increased and correlated with clinical disease activity. Gal-9 levels remained increased despite a significant reduction in the disease activity score in patients with early RA. The in vitro neutralization of Gal-9 decreased both MCP-1 and IL-6 production in an inflammatory subset of RA FLSs. Collectively these findings indicate that the persistent overexpression of Gal-9 in RA may modulate synovial FLS activities and could be involved in the maintenance of subclinical disease activity in RA.

## 1. Introduction

Rheumatoid arthritis (RA) is the most common systemic immune-mediated inflammatory joint disease. RA is characterized by a chronic joint inflammation, ultimately leading to cartilage and bone degradation, joint deformities and loss of function if the disease is not sufficiently controlled. The rheumatoid synovial membrane is characterized by extracellular matrix expansion, which harbors an abundant cell infiltrate mainly composed of fibroblasts, macrophages and inflammatory lymphocytes [1]. Within the past two decades, the treatment of RA has been radically improved due to the introduction of earlier onset and intensive therapies, plus the development of biological immunosuppressive therapies predominantly targeting leukocytes and their cytokine production [2,3]. However, up to 30% of patients with RA fail to achieve adequate disease control on these treatments [4]. Thus, there is an unmet need for elucidating the local modulators of the inflamed microenvironment and, hence identifing potential new treatment targets [5,6,7]. 

Fibroblast-like synovial cells (FLSs) play an important role in the inflamed synovial environment and have been reported to respond poorly to synthetic and biologic disease-modifying anti-rheumatic drugs (DMARDs) [8,9,10]. Although FLSs are considered to be important drivers of RA pathology, due to their interplay with immune and endothelial cells, little is known about the mechanisms important for this interplay [11,12]. Studies using single-cell resolution have led to the identification of RA synovial fibroblast subsets with specific inflammatory properties. Thus, a subset of fibroblasts (CD34^−^PDPN^+^THY1^+^), which produce high levels of MCP-1 and IL-6, is particularly prevalent in the inflamed synovium of RA patients with a poor response to currently approved anti-rheumatic drugs [1,13,14,15]. 

Polymorphisms in the gene encoding the galectin-9 (Gal-9) are associated with RA development [16], and Gal-9 has been reported to be elevated in the serum of RA patients, compared with healthy controls [16,17]. Elevated levels of Gal-9 have also been reported in both the synovial fluid and in the inflamed synovial tissue of patients with RA [18]. In RA, Gal-9 appears to be capable of both decreasing and increasing the viability and inflammatory activity of synovial FLSs, dependent on its intra- or extracellular localization [18,19]. Furthermore, galectins, as multivalent proteins, are proposed to anchor cells, enzymes or cytokines to the joint through adherence to extracellular matrix proteins, displayed in the inflamed RA synovium [20,21]. Gal-9 contains two carbohydrate recognition domains (CRDs) with an affinity for β-galactosides, a motif characteristic of galectins, but the glycan binding specificity of Gal-9 is different from other galectin family members [22,23,24]. 

In the present investigation, we aimed to study circulating Gal-9 in patients with early RA (eRA) before treatment and established RA (esRA), and to study possible associations with disease activity and radiographic outcome following a two-year treat-to-target protocol. In addition, we investigated the mechanisms of action of Gal-9 on FLSs, based on mononuclear cell cultures from the synovial membrane and fluid collected from patients with OA and RA.

## 2. Materials and Methods

### 2.1. Patient Material

Plasma samples were collected serially (0 and 3 months) from 98 randomly selected patients among those enrolled in the Danish multicenter study *Optimized treatment in Early RA [OPERA; registered at ClinicalTrials.gov (NCT00660647)].* Details of the study design have been previously published [25,26]. Briefly, treatment-naïve patients with newly discovered RA were randomized to a conventional MTX treatment, plus placebo (DMARD + PLA) or MTX, in combination with adalimumab, an anti-TNFα agent (DMARD + ADA). At entry (baseline), patients had a disease duration (from first persistently swollen joint) of <6 months (average 3 months) and had moderate to severe RA, defined as a 28-joint DAS with CRP (DAS28-CRP) (four variables, CRP-based) >3.2. Radiographic progression was assessed by the delta total Sharp score (TSS). Disease activity was evaluated by the simple disease activity index (SDAI), Clinical disease activity index (CDAI), and DAS28-CRP. DAS28-CRP is based on CRP, the number of swollen (28 and 40) and tender joints (28 and 40), and a physician’s global assessment of the disease activity. The baseline DAS28-CRP was 5.75 (CI 5.54–5.96), which was reduced to 2.1 (1.8–3.2) after 3 months in both treatment arms and to 2.0 (1.8–2.7) after 24 months (Table 1). Clinical and laboratory data were acquired at the same time as the plasma samples were collected, but also after 24 months wherefrom, plasma samples were not assessed. 

In addition, we obtained a paired set of plasma (P) and synovial fluid (SF) from patients with established RA (*n* = 18) in a single-center cross-sectional study at the outpatient clinic at Aarhus University Hospital. The patients enrolled had, at the time of therapeutic arthrocentesis, a DAS28-CRP of 5.0 (CI 3.2–5.5) (Table 1). These patients were treated in accordance with current treatment guidelines for RA [27].

The RA patients ©ded in this study were all >17 years of age and fulfilled the ACR 1987 revised criteria for RA. 

Plasma samples from healthy controls (*n* = 48, HC) were obtained from the Danish Blood Bank, Aarhus University Hospital. The control samples were obtained, anonymized, from a blood bank. The exact number of patients in each experiment is provided in the figure legends. Synovial tissue fragments were collected from patients with established RA, undergoing arthroplasty (*n* = 6).

Osteoarthritis FLSs were obtained from OA patients with knee involvement undergoing joint replacement surgery (*n* = 5). OA patients were diagnosed with knee OA with a Kellgren–Lawrence score of 2 to 4, fulfilling the 1985 criteria of the American Rheumatism Association.

### 2.2. Sample Handling

Plasma samples were EDTA-stabilized and stored at −80 °C. The PBMCs and SFMCs from established RA patients were isolated immediately after sample acquisition by conventional Ficoll-Paque (GE Healthcare, Chicago, IL, USA) density-gradient centrifugation and were cryopreserved at −135 °C until the time of analysis. 

### 2.3. Culture Conditions

In vitro cultures of human RA synovial fluid cells.

### 2.4. RA SFMCs In Vitro Cultures 

RA SFMC 48-h cultures are a cellular-mixed culture of mainly lymphocytes and monocytes [28,29]. RA SFMCs were thawed and cultured in Dulbecco’s modified Eagle’s medium (DMEM), supplemented with 10% fetal calf serum (FCS), 1% penicillin, 1% streptomycin, 1% glutamine and HEPES 10 mM (Gibco) (culture media) at a density of 1 × 10^6^ cells/mL. The cells were cultured at 37 °C in a humidified incubator with 5% CO_2_ for 48 h.

### 2.5. RA and OA FLSs In Vitro Cultures

RA FLSs were grown from SFMCs, as previously described [30]. In short, when the cell layer reached >70% confluence, the FLSs were passaged using trypsin/EDTA (Invitrogen) treatment and used for analyses at passage 4–5. OA FLSs were isolated from synovial tissue, as previously described [31]. RA and OA FLSs were seeded at 2 × 10^4^ cells/mL in culture medium. When the cell layer reached >70% confluence, the media was changed and the compounds were added and cultured for 48 h.

### 2.6. Anti-Gal-9 Antibody Treatment of FLS Cultures

FLSs, purified, form SFMCs, activated in situ, spontaneously secreting MCP-1 and IL-6, were treated with neutralizing murine anti-human Gal-9 IgG (Gal-Nab2, 10 µg/mL) [32], a murine isotype matched IgG as control (Jackson, 015-000-002, 10 µg/mL) or hydrocortisone (HyC, 1 µg/mL). As a positive control, lipopolysaccharide (LPS, 100 ng/mL) was included. The cells and supernatants were harvested after a 2-step centrifugation (450 g/10 min/RT) separation, as previously described [30]. The cells were directly processed for flow cytometry analysis and the cell-free supernatants were stored at −80 °C until analysis. 

### 2.7. RA Autologous FLS and PBMC In Vitro Co-Cultures

The cross-talk between pathological fibroblasts and autologous leukocytes was investigated using an in vitro autologous co-culture of RA FLSs and PBMCs [30]. The FLSs were grown as described above. When the FLS cell layer reached >70% confluence, autologous PBMCs were added at 7.5 × 10^5^ cells/mL and anti-Gal-9 treatment or controls were added, as described above. Again, LPS 100 ng/mL was added as a positive control and the cells were co-cultured for 48 h. Cells were directly processed for flow cytometry analyses and the cell-free supernatants were stored at −80 °C until analysis. 

### 2.8. ELISA

Gal-9 levels in plasma and synovial fluid were quantified using a commercial Gal-9 Quantikine ELISA kit (Cat no. DGAL90 R&D Systems, Minneapolis, MN, USA) according to the manufacturer’s instructions with the following modifications: plasma samples were pretreated to avoid interference with heterophilic antibodies [33] and all wells were additionally blocked by Synblock^®^ (Cat no. BUF034B Bio-Rad, Hercules, CA, USA) to reduce unspecific binding. MCP-1 and IL-6 levels in supernatants were assessed by the ELISA MAX™ Deluxe Set Human MCP-1/CCL2 and ELISA MAX™ Deluxe Set Human IL-6 (Biolegend, San Diego, CA, USA), respectively. Values below the detection limit were assigned the same value as the detection limit. A Thermo Scientific Multiskan GO reader was used to acquire the OD of the wells at 450 nm with a reference reading at 570 nm. 

### 2.9. Flow Cytometry

FLSs were blocked with 0.1 mg/mL mouse IgG (Jackson ImmunoReseach, West Grove, PA, USA) and 0.1 mg/mL rat IgG (Jackson ImmunoReseach) for 15 min and then surface stained with CD34-PerCP-eFlour710 (Clone: 4H11, Thermo Fisher Scientific, Waltham, MA, USA), CD45-APC-Cy7 (Clone: HI3, Biolegend), THY-1-PE-Cy7 (Clone: 5E10, Biolegend, San Diego, CA, USA), PDPN-PE (Clone: NZ-1.3, eBioscience, Thermo Fisher), and fixable LIVE/DEAD nIR Dead Cell Stain Kit (Life Technologies, Thermo Fisher Scientific). Data were acquired by NovoCyte Quanteon^®^ (ACEA Bioscience Inc., San Diego, CA, USA) within 24 h of surface stain and processed in FlowJo (FlowJo software version 10.5.3). The spectral overlap was compensated using antibody-coated beads (eBioscience, San Diego, CA, USA). Gating was performed on live cells using fluorescence minus one (FMO). 

### 2.10. Immunofluorescence of RA Synovial Tissue

Paraffin-embedded and formalin-fixed RA synovial tissue samples were cut (5 µm sections) and subjected to standard deparaffinization and rehydration steps prior to antigen retrieval by heat. Sections were blocked for unspecific binding (PBS supplemented with 0.5 % BSA and 10 % donkey serum). Gal-9 expression was evaluated using the rabbit anti-human Gal-9 1:50 (Cat. PA5-32252; ThermoFisher, Waltham, MA USA) antibody followed by an Alexa 647-conjugated secondary anti-rabbit antibody 1:50 (Cat. Jackson 711-606-152). Primary antibodies were omitted for the negative controls, otherwise performed in the same way. Slides were mounted in Prolong Gold Antifade Mounting media with DAPI (Life Technologies, Carlsbad, CA, USA). Representative images were acquired using an Olympus VS120 slide scanner at 10× and 40× and processed using the imaging software packed CellSens Standard (Olympus, Tokyo, Japan). 

### 2.11. Cell Viability of In Vitro Co-Cultures

The cell viability of FLS monocultures and FLS and PBMC co-cultures was determined after 48 h of culturing by the commercially available Cell viability and proliferation assay (MTT) (Cat. No. 11-465-007-001, Roche, Mannheim, Germany), according to the manufacturer’s instructions. A Thermo Scientific Multiskan GO reader was used to acquire the optical density of the wells at 570 nm, with a reference reading at 750 nm. 

### 2.12. Microarray Analysis

The normo-glycosylated extracellular matrix proteins (ECMs) were printed on Oncyte^®^ nitrocellulose film slides (Grace Bio-Labs, Bend, OR, USA) using a sciFLEXARRAYER S11 (Scienion, Berlin, Germany). Collagen I (234138, Sigma-Aldrich, Saint Louis, MO, USA), Elastin (324751, Sigma-Aldrich), Collagen II (CC052, Sigma-Aldrich), Laminin (AG56P, Sigma-Aldrich), Thrombospondin (605225, Sigma-Aldrich), Fibronectin (341635, Sigma-Aldrich), Collagen IV (CC076, Sigma-Aldrich), Collagen III (CC054, Sigma-Aldrich), Vitronectin (CC080, Sigma-Aldrich), IgM bulk, IgG bulk, IgG FC (Sigma-Aldrich), and Chicken collagen II (C9301, Sigma-Aldrich), purchased from the various vendors mentioned. Collagen II was solubilized by digestion in 0.25% acetic acid (pH 3.1) over several hours at 2–8 °C, with occasional vortexing and sonicating. All proteins were dissolved in PBS and printed at a concentration of 100 μg/mL in replicates of 4 spots/protein. Following printing, the slides were incubated overnight at 4 °C. Then, the slides were treated with Super G Plus™ Protein Preservative (Grace Bio-Labs), as per the manufacturer’s recommendation, to block from non-specific binding. The slides were then stored at −20 °C in an airtight tube container until time of use. The printed proteins were verified independently with corresponding antibodies, data not shown.

Regarding the Gal-9 assay, after rehydration using TSM buffer (20 mM Tris–HCl, 150 mM sodium chloride, 0.2 mM calcium chloride, and 0.2 mM magnesium chloride), the microarray slides were probed with or without 5 μg/mL rhGal-9 (9064-GA, R&D Systems). The bound Gal-9 was detected using goat anti-Gal-9 5 μg/mL (AF2045, R&D Systems), followed by fluorescent anti-goat IgG-633 (H + L) 5 μg/mL (A21086, Thermo Fischer). In a control set-up, Gal-9 was omitted from the set-up otherwise performed in the same way. The binding signals were directly quantified after the incubation and slides were scanned with a GenePix 4300A microarray scanner (Molecular Devices, Sunnyvale, CA, USA), photomultiplier (PMT): 450, Laser Power (LP): 25. Spot-based signal intensities were quantified using GenePix Pro 7 (Molecular Devices, San Jose, CA, USA). The raw data from the software was processed using Microsoft Excel to obtain the background-subtracted mean relative fluorescence intensity (RFU) for the four replicates, as previously described [34].

### 2.13. Statistics

Statistical analyses and graphs were performed using GraphPad Prism 7 for Mac (GraphPad Software, La Jolla, CA, USA). The amount of MCP-1 secreted in the cultures was calculated as ratios comparing stimulated cells with untreated cells, due to data normalization and donor variations. The differences were compared with the corresponding controls. Data were analyzed with a paired or unpaired *t*-test. Correlations were studied using Spearman’s *ρ* test. In all tests, the level of significance was a two-sided *p* value of less than 0.05. Graphics are presented as the mean, with standard error of the mean (SEM) if not otherwise stated. 

### 2.14. Ethics

The OPERA study was approved by the Danish Medical Agency (2612-3393), the Danish Data Protection Agency (2007-41-0072), and the Regional Ethics Committee (1-10-72-82-15).

The use of human cells from the INART biobank was approved by the Danish Data Protection agency and the Ethics Committee, protocol number: 20-121-329. The subjects’ informed and written consent was obtained, according to the Declaration of Helsinki.

## 3. Results

### 3.1. Increased Gal-9 Levels in Plasma and Synovial Fluids in Rheumatoid Arthritis

At baseline, untreated patients diagnosed with RA had significantly higher plasma levels of Gal-9 compared with HC (Figure 1A). Treatment, using a treat-to-target approach, resulted in a significant reduction in the disease activity score DAS28CRP, after 3 and 24 months of treatment (Table 1). By contrast, Gal-9 plasma levels remained, on average, unchanged over the first 3-month treatment period (Figure 1A), and there was no significant difference between the two treatment regimens (Figure 1B). At baseline, Gal-9 correlated with the number of swollen joints (rho = 0.344, *p* = 0.001), CDAI (rho = 0.24, *p* = 0.019) and SDAI (rho = 0.23, *p* = 0.024), but not with DAS28CRP or deltaTSS. Furthermore, baseline levels of Gal-9 did not correlate with long-term disease activity, measured by DAS28CRP, and radiographic progression, measured by deltaTSS; these were evaluated after 24 months of treatment. 

Plasma and synovial fluid from patients with established RA and disease flare were also examined for Gal-9. In the plasma of esRA patients, levels of Gal-9 were significantly higher than for HC, but similar to the levels measured in eRA patients. In esRA, the Gal-9 levels in the synovial fluid were more than 8 times higher than in plasma (Figure 1A). We also detected the broad expression of Gal-9 in the inflamed synovial tissue from RA patients, especially in the sublining layer and surrounding vessels (Figure 1C). 

### 3.2. Gal-9 Binds to Human Laminin, Fibronectin, and Thrombospondin and IgG

As galectins have been reported to bind to glycosylated ECM components, displayed in the inflamed RA joint, we examined whether Gal-9 had a similar ability, by using human normo-glycosylated ECM components printed on microarray slides (Figure 2). Gal-9 exhibited a relatively high binding to human laminin, fibronectin, and thrombospondin; this is compared to lower to no binding to a number of other ECM components, including collagens, elastin and vitronectin. In addition, Gal-9 also bound human bulk IgG with a preference for the Fc-part, but not to IgM (Figure 2). 

### 3.3. Anti-Gal-9 Antibodies Inhibit Cytokine Production by Fibroblast-Like Synoviocytes

Since Gal-9 has been closely linked with the activation and survival of FLSs in RA, we studied the effect of neutralizing Gal-9 in inflammatory RA FLSs. Addition of a neutralizing Gal-9 antibody elicited a 40% decrease in MCP-1 secretion (*p* < 0.001) as well as a 30% decrease in IL-6 secretion (*p* < 0.05) in these RA FLS monocultures (Figure 3A and Appendix A). FLSs from patients with knee OA were included as disease controls. In these cultures, the addition of anti-Gal-9 antibodies also decreased the production of both MCP-1 (*p* < 0.01) and IL-6 (*p* < 0.05) (Figure 3B). However, despite comparable percentage reductions in MCP-1 and IL-6 secretion, RA FLSs tended to have a higher secretion of both MCP-1 (30%) and IL-6 (60%), compared with OA FLSs (Appendix A). The MCP-1 and IL-6 reduction was not a result of a reduction in the fraction of inflammatory FLSs (CD34^−^PDPN^+^THY1^+^), nor was it due to decreased viability and proliferation (Figure 3C).

The effect of neutralizing Gal-9, also in a co-culture between FLSs and autologous PBMCs, was then evaluated. Again, the neutralization of Gal-9 led to a 40% reduction in both MCP-1 and IL-6 (*p* < 0.05, *p* < 0.001, respectively) (Figure 3D). In all instances, as expected, LPS induced a > 3-fold increase in cytokine production (Appendix A). The inhibitory effect of neutralizing Gal-9 on MCP-1 and IL-6 secretion was not seen in SFMC cultures without differentiated FLSs (Appendix A). 

## 4. Discussion

These results demonstrate that circulating Gal-9 levels are increased in early RA, but are unresponsive to targeted synovitis-suppressive treatments while associating with pretreatment disease activity. The addition of anti-gal-9 antibodies to monocultures of FLSs, or co-cultures of FLSs plus autologous PBMCs, significantly diminished the secretion of MCP-1 and IL-6 in both settings. Collectively, the results suggest that the local increase in and effects of Gal-9 are associated with disease activity in RA, and that Gal-9 regulates pathogenic FLSs.

Gal-9 plasma levels remained elevated throughout the first three months of treatment, despite significant clinical improvements in both treatment arms. This contrasts with our earlier reports on markers of T-cell activation, which decreased following treatment initiation [35]. These observations may imply that the present treatment protocol, which comprises both csDMARD as mono-therapy and in combination with a bDMARD, does not directly target the mechanisms underlying the increased Gal-9 levels in eRA, despite earlier associations between Gal-9 and TNFα in RA [36]. The finding that Gal-9 levels were essentially similar between eRA baseline values vs. esRA on-treatment levels further supports this. That plasma Gal-9 levels correlated strongly with the number of swollen joints and were highly increased in the synovial fluid, further points to the role of Gal-9 in local RA pathology. We did not see an association between Gal-9 plasma levels and the development of joint erosions. We speculate that the reason for this may be the limited general increase in bone erosions in the OPERA cohort [26] and the limited resolution of conventional X-Ray, compared with MR or CT. 

In order to study local Gal-9 responses in further detail, we focused on the possible impact of Gal-9 on CD34^−^PDPN^+^THY1^+^-expressing FLSs, since these cells have emerged as a currently untargeted proinflammatory cellular subset that is locally upregulated in the sublining layer of the RA synovial membrane [13,14,37]. The factors regulating the production of proinflammatory cytokines in this FLS subset are currently unknown. Notably, however, we observed that the addition of a neutralizing Gal-9 antibody to FLSs in mono- or co-cultures with autologous PBMCs was associated with up to a 40% reduction in both MCP-1 and IL-6 production by FLSs. This is a new and potentially pertinent observation because the FLS inflammatory activity is resistant to current treatment regimens, both in human RA and in mouse models [10,38]. Based on our finding of a markedly increased synovial fluid:plasma Gal-9 ratio, it seems plausible that the inflamed RA synovial membrane is a major source of increased Gal-9 in its systemic circulation in RA patients. 

Our observation that Gal-9 binds to human components of the ECMs, including laminin, fibronectin and thrombospondin, accords with previous reports on another galectin, Gal-3 [39,40]. That Gal-9 binds to laminin, fibronectin and thrombospondin indicates that these ECMs contain available complex-branched N-glycans [34] and that the collagens acquired from non-inflamed tissue do not contain sufficiently available complex-branched N-glycans. 

By this mechanism, galectins may eventually be trapped within the joint inflammatory niche and subsequently exert their regulatory actions in a targeted manner, according to the composition of the local ECM microenvironment. This is supported by the finding that Gal-9 is particularly prevalent in the sublining layer of the synovial membrane (Figure 1C). All the ECMs printed were chosen based on their known highly normo-glycosylated state containing complex N-glycans. However, the ECMs were commercially acquired from non-inflamed tissue and, therefore, the possible changes to these proteins, mediated by the inflamed microenvironment [41], may alter the specific binding in the inflamed joint. Even though Gal-9 does not directly bind IgM, the presence of IgM-RF will have the ability to localize within the same area, thereby possibly amplifying local inflammation. Conceptually, then, the binding of Gal-9 to exposed ECMs may further activate resident pathological FLSs, thereby contributing to local tissue damage.

The neutralization of Gal-9 resulted in a decrease in cytokine production in FLS-containing cultures, whereas no effect was seen in RA SFMC cultures. Thus, Gal-9 neutralization does not work in the way observed with the majority of existing biological DMARDs [30]. This is likely due to the cellular composition of these models, as the SFMC cultures are rich in monocytes and lymphocytes [28], but are devoid of fully differentiated stromal cells. Thus, our findings indicate that the neutralization of Gal-9 does not robustly affect the production of MCP-1 and IL-6 in a lymphocyte and monocyte-dominated environment without FLSs, even though galectins are known to be involved in a variety of immune cell modulations in arthritis [42,43,44]. 

Interestingly, the response patterns of OA FLSs were similar to that observed in RA, indicating that, despite differences in pathology, inflammatory FLSs are also present in OA tissue [45,46]. The fact that galectin inhibitors are capable of decreasing the cytokine production in both these monocultures indicates that these fibroblasts are, to some extent, capable of self-activation without additional stimuli from immune cells in both diseases. Thus, it is likely that these inflammatory FLSs maintain their intrinsic activity and cytokine production, despite suppression of leukocytes by the various DMARDS. 

Based on the present study, the neutralization of Gal-9 in human RA may be a supplement to a selected group of DMARD unresponsive patients, as Gal-9 plays a central role in the production of proinflammatory cytokines produced by pathological FLSs. Thus, Gal-9 neutralization could potentially be effective as a supplement to anti-TNFα, treatment with the purpose of also targeting the pathological FLSs in RA [47], akin to specific Gal-3 inhibitors [39].

To study the role of local Gal-9, we utilized three different ex vivo RA-cultured systems. These cultures consisted of in situ preconditioned mononuclear cells and FLSs grown from cells harvested from the synovial fluid. There are several advantages to such a study design, as these models are driven by the spontaneous endogenous production of multiple pro-inflammatory factors, without the addition of any exogenous stimulation. Even though it is not possible to translate results directly to an in vivo setting, this experimental design offers the possibility of isolating the different components of arthritis pathobiology and evaluating them separately; however, an unavoidable consequence is the lack of some of the complexity of in vivo experiments. 

In conclusion, these findings provide additional mechanistic evidence for the role of Gal-9 in local RA joint pathology. The systemic Gal-9 levels correlated with baseline disease activity but remained unaffected during csDMARD and anti-TNFα treatments, despite decreasing disease activity. The neutralization of Gal-9 decreased the secretion of MCP-1 and IL-6 in arthritis synovial cultures, dominated by FLSs. Collectively, these observations support that Gal-9 is implicated in the RA pathogenetic pathway by regulating FLS activities within the rheumatoid synovial membrane. Targeting Gal-9 by neutralizing antibodies may emerge as a useful therapeutic adjunct in patients who do not respond adequately to current intensive synovitis-suppressive strategies.

## Figures and Tables

**Figure 1 cells-12-00327-f001:**
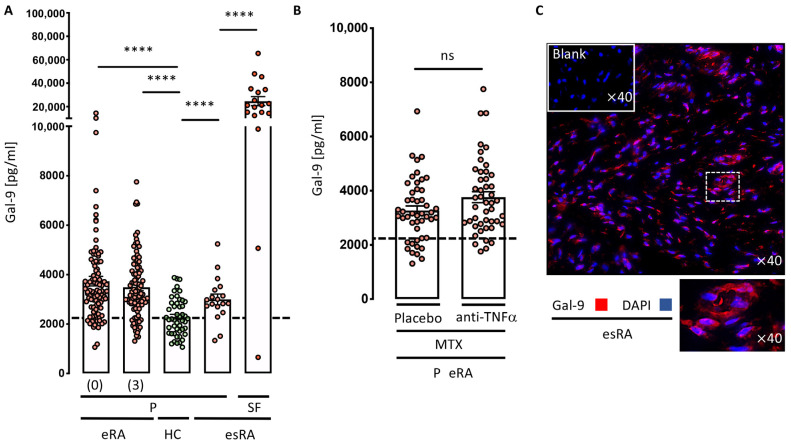
Gal-9 levels in plasma (P) or synovial fluid (SF). (**A**) Samples from 98 early RA (eRA) patients were analyzed at baseline (0) or after 3 months treatment (3) and compared to 48 healthy controls (HC). Paired samples of plasma and SF from 18 established RA (esRA) patients were also analyzed. Dashed line represents mean HC Gal-9 levels. (**B**) Gal-9 levels in plasma after 3 months of the two treatment regimens (MTX + placebo, *n* = 50) and (MTX + anti-TNF, *n* = 48). Dashed line represents mean HC Gal-9 levels. (**C**) Representative synovial tissue expression of Gal-9 and DAPI (*n* = 6). Differences were analyzed using the either paired or unpaired *t*-tests to compare two groups. Bars indicate Mean ± SEM. **** *p* < 0.0001. ns: not significant.

**Figure 2 cells-12-00327-f002:**
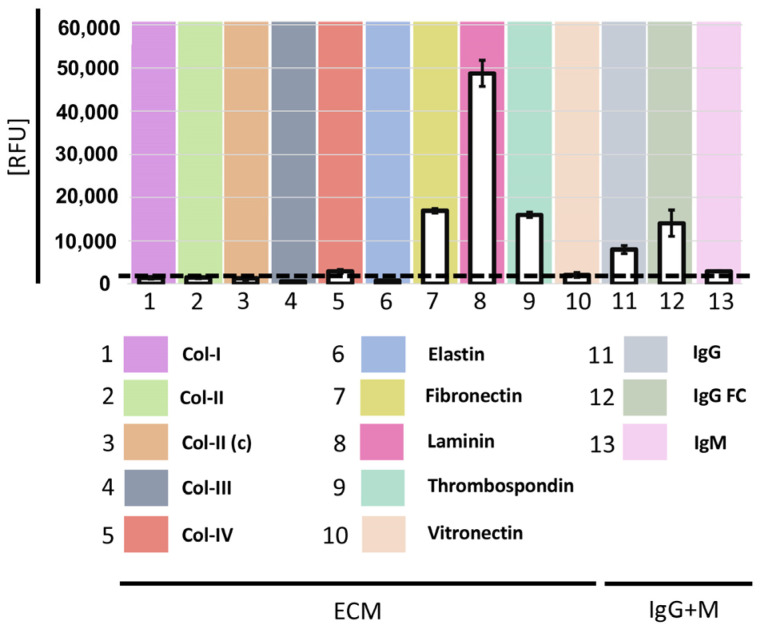
Binding of Gal-9 (5 μg/mL) to a selection of normo-glycosylated ECM proteins in a microarray format. Results are expressed as relative fluorescence units (RFU) by averaging the background-subtracted fluorescence signals of four replicate spots, with error bars representing the standard deviation among the four values.

**Figure 3 cells-12-00327-f003:**
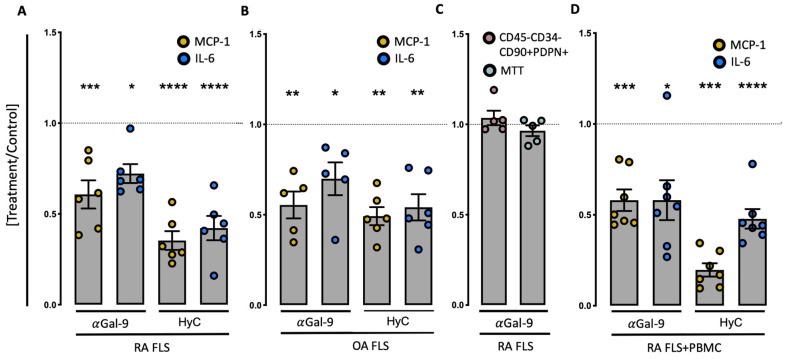
The secretion of MCP-1 and IL-6. Results are expressed as ratios of the cytokine concentrations in neutralizing Gal-9, (Gal-Nab2, 10 µg/mL) and Hydrocortisone (HyC) (1 µg/mL)-treated monocultures of fibroblast-like synovial cells (FLS), divided by the concentration in control cultures, cultured for 48 h (RA *n* = 6), (OA (HyC, *n* = 6), (Gal-Nab2, *n* = 5)). (**C**) Evaluation of pathological FLSs and viability (MTT values) of the anti-Gal-9 treated cells in (**A**) and (**B**). FLSs incubated with the neutralizing anti-Gal-9 antibody. Data are expressed as ratios between the proportion of pathological FLSs (CD45-, CD34-, CD90+, and PDPN+) out of all CD45-FLS incubation with the neutralizing anti-Gal-9 antibody or isotype control IgG (*n* = 5), or the ratio between mTT ODs of all FLSs incubated with anti-Gal-9 or isotype control IgG (*n* = 5); in both set-ups the cells were cultured for 48 h. (**D**) Secretion of MCP-1 and IL-6 from 48 h RA FLSs + PBMC co-cultures. Data are expressed as ratios between the cytokine concentrations in neutralizing Gal-9, (Gal-Nab2, 10 µg/mL) and Hydrocortisone (HyC) (1 µg/mL)-treated co-cultures of fibroblast-like synovial cells (FLSs) + autologous PBMCs, divided by the concentration in control-treated cultures, cultured for 48 h (RA *n* = 7). Differences were analyzed using the paired *t*-tests to compare two groups. Untreated RA co-cultures of FLSs + autologous PBMCs produced, on average, 30,055 ± 4063 pg/mL MCP-1 and 44,624 ±5006 pg/mL IL-6 (Mean ± SEM). Bars indicate mean ± SEM. * *p* < 0.05, ** *p* < 0.01, *** *p* < 0.001, **** *p* < 0.0001.

**Table 1 cells-12-00327-t001:** Patient characteristics in treatment-naïve early rheumatoid arthritis (RA) (*n* = 98), established RA (*n* = 18), and healthy controls (HC) (*n* = 48). Data are expressed as median with IQR range unless otherwise indicated. ADA Adalimumab, CCP Cyclic citrullinated peptide, DAS28CRP disease activity score 28 based on C-reactive protein, IgM-RF IgM rheumatoid factor.

Patient Characteristics	Early RA (*n* = 98)			Established RA (*n* = 18)	HC (*n* = 48)
Time since diagnosis (months)	0	3	24	84 (12–288)	
Age (years)	56 (46–65)	-	-	47 (37–61)	49 (45–58)
Gender (% female)	73	-	-	70	57
Baseline characteristics					
IgM-RF (% positive)	68	-	-	39	-
Anti-CCP antibody (% positive)	57	-	-	46	-
Treated with ADA (%)	49	-	-	8	-
Disease activity					
DAS28CRP (0–10)	5.7 (5.1–6.4)	2.1 (1.8–3.2)	2.0 (1.8–2.7)	5.0 (3.2–5.5)	-

() illustrates that the numbers in (xx-xxx) in the last part of the table is Median + IQR: (Median with IQR.)

## Data Availability

The raw data supporting the conclusions of this article will be made available by the authors, without undue reservation.

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
