# Peer review of "Increased Galectin-9 Levels Correlate with Disease Activity in Patients with DMARD-Naïve Rheumatoid Arthritis and Modulate the Secretion of MCP-1 and IL-6 from Synovial Fibroblasts"

_cells, 2023, doi:10.3390/cells12020327_

Round 1

Reviewer 1 Report

The authors describe association with Gal-9 levels with disease activity in DMARD-naive RA and modulation of MCP-1 and IL-6 from synovial fibroblasts harvested from RA patients. The result of study is convincing and of interesting to readers, however, I have several concerns. Specific comments are listed below: 

1. Not only elevated levels of Gal-9 in RA patients, association between Gal-9 levels and several cytokines (TNF-alpha and IL-6) also have been reported in established RA patients in previous report (Matsumoto, PlosOne 2020). This study showed strong correlation with TNF-alpha levels and Gal-9 levels in established RA, rather than early RA.  Figure 1B showed Gal-9 levels were not decreased in TNF-alpha treated RA patients compared to Placebo+MTX patients, hence, what is the reason for this?  Please add the discussion.  

2. Figure1: please spell out eRA and esRA at first time (may be confusing). 

3.  A previous report has described that Gal-9 levels were strongly correlated with anti-CCP titers especially in patients with high APCA-positive RA (Fujita, Arthritis Res Ther 2020). The binding of Gal-9 to ECM proteins (which are associated with ACPA) shown in Figure 2 may be in accordance with previous studies. What imply this? Please add discussion in that point.  

4. Schematic presentation of this study's findings can be helpful for readers to understand what is new and important points.

Reviewer 2 Report

As one of many additional studies of OPERA , the authors present here a translational study investigating clinical associations and the overall role of galactin-9 in RA.

The manuscript is well written and understandable. 

One major points needs further consideration. 

You write that Gal-9 is associated with some disease activity in RA but only in untreated patients. In my opinion this needs further clarification and some further analysis. 

What about the change of Gal-9 before and after treatment? Does it only change in the ones responding to treatment? Do you have correlating US and/or MRI examination of the hands which would allow to make associations with subclinical activity (i think in OPERA MRI was part of the protocol!?). 

What about patients treated with Glucocorticoids? SF react very fast to GC, so there at least you should be able to see a change of Gal-9.

The second measurement of Gal-9 was after three months, as you have serum after 12 month as well, i suggest to measure Gal-9 there again. Maybe Gal-9 production diminishes in sustained response. 

Reviewer 3 Report

The authors note that it seems plausible that the inflamed RA synovial membrane is a major source of increased Gal-9 in the systemic circulation in RA patients (Line 390). I feel that this conclusion is adequately supported by the experimental results that are presented, and I believe this finding will almost certainly be of interest to the readership of your journal. Chasing down the role of galectin-9 in rheumatoid arthritis is important as it could lead to additional therapeutic strategies. For these reasons, I definitely recommend publication of this article.

Additional discussion is needed regarding the outcome of the microarray binding experiment:

The authors should provide more discussion of why they think galectin-9 binds to some ECM proteins such as laminin but not collagens and some other proteins studied. Were these proteins chosen because the authors thought galectin-9 might bind to them, but then they didn’t? Are there fewer galactosides displayed on the proteins that are not bound well by galectin-9? Are some glycans more prevalent in laminin than in any of the other ECM proteins? Is the spacing of galactosides different in laminin? What ligand in laminin is galectin-9 most likely binding? The authors note that galectin-3 and galectin-9 are reported to have differing carbohydrate binding affinities, but they indicate that their array binding specificities observed for galectin-9 in this publication are in accord with previously reported results for galectin-3. The results of the microarray binding experiment should be more completely explained.

Other small issues to address prior to publication:

Figure 1. I think the dashed line represents the level of gal-9 in healthy control at (0) in both A and B, but this should be specified in the figure legend. C Are these cells at 0 or 3 months? Are they treated as in B or untreated as in A?

Line 366 should be regulates, not regulate

Line 370 should be in, not I

Round 2

Reviewer 2 Report

Thank you. No further comments.